

# Brief communication: Improving lake ice modeling in ORCHIDEE-FLake model using MODIS albedo data

Zacharie Titus[1,2], Amélie Cuynet[1], Elodie Salmon[1] and Catherine Ottlé[1]

[1]Laboratoire des Sciences du Climat et de l'Environnement, IPSL, CEA-CNRS-Université Paris-Saclay, Orme des Merisiers, Gif-sur-Yvette, 91190, France

[2]Laboratoire de Météorologie Dynamique, IPSL, Sorbonne Université, Institut Polytechnique de Paris, ENS, Palaiseau, 91120, France.

*Correspondence to*: Catherine Ottlé (catherine.ottle@lsce.ipsl.fr)

**Abstract.** The Flake lake model embedded in the ORCHIDEE land surface model was recently updated to better represent winter ice cover. MODIS albedo data and the Great Lakes Ice Cover fraction dataset over the Laurentian Great Lakes were used to calibrate and validate a new parameterization of the lake albedo accounting for a partial ice cover fraction. The results show large improvements in the simulation of the ice phenology of 200 lakes of various sizes reported in the Global Lake and River Phenology database. The agreement with the observations is improved for all lake size categories, with the largest and deepest lakes showing larger error reductions on the duration of the ice cover period. This study highlights the importance of considering partial ice cover to correctly model lake albedo in cold regions and thus to simulate realistic mass and energy exchanges at the land-atmosphere interface.

## 1 Introduction

The role of lakes on the atmosphere-surface water and energy exchanges at local, regional, and global scales has been demonstrated in various works (Tranvik et al., 2009, Williamson et al., 2009, Woolway et al., 2020, Huang et al., 2023, to cite but a few). Therefore, their representation in climate and weather prediction models becomes a necessity to improve numerical predictions. Lakes present specific physical properties compared to their surroundings (lower albedo and surface roughness, larger heat capacity and thermal conductance) that impact the energy exchanges with the atmosphere. Lakes are also a significant source of water, carbon and methane whose impact on climate is still not precisely quantified (Johnson et al., 2022, Lauerwald et al., 2023). Moreover, in cold regions, lakes can freeze and store snow, with large impacts on the seasonal evolution of water and energy surface fluxes, but also on carbon exchanges, as reported by several authors because of the accumulation of gases under the ice cover that can be released abruptly when the ice melts (Mammarella et al., 2015, Denfeld, et al., 2018). The representation of freezing processes is, therefore, key for modeling the dynamics of the atmosphere thermodynamics and greenhouse gases (carbon and methane) budgets, especially in the present context of climate warming. Indeed, long time series of observations have highlighted the impacts of the combined effects of air temperature increase, but also of volume variations on lake temperature and thermal regime, as well as on the loss of winter ice cover (O'Reilly et al., 2015, Woolway et al., 2020, Huang et al., 2023, Sharma et al., 2015). To better understand these connected processes, and better predict future trends, climate models need to develop lake modules able to correctly represent water and energy budgets as well as the carbon cycle. Such developments have been ongoing in the climate modeling community for several years, with

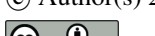



major works demonstrating the benefits of representing lakes at the global scale in Numerical Weather Prediction and climate models (e.g., Dutra et al., 2010, Balsamo et al., 2012, Le Moigne et al., 2016).

In these large-scale models, the representation of lakes is generally done simplistically, based on 1D column models which do not represent the spatial variability of the surface exchanges linked to the varying bathymetry. The same applies to the freezing processes and the fact that ice cover is considered uniform laterally. However, its formation can take several days /weeks and
may never be completed for some large and deep lakes.  Moreover, because of the lake bathymetry and the sometimes large spatial variations in lake depth, the mass and energy transfers can present very heterogeneous features, so the ice cover and the resulting impacts on surface fluxes, as shown for methane emissions on the Tibetan lakes by Lang et al. 2018.

The modeling of ice conditions is an important challenge in such models for several reasons such as that the processes involved are strongly nonlinear, because free water and ice show very different characteristics, and also exhibit high spatial and temporal
variability. Scale issues must, therefore, be parameterized to account for these variabilities. Lake temperature is the result of the energy budget, which is strongly dependent on the surface albedo which determines the amount of solar energy absorbed at the surface. When the air temperature falls below 0°C, the lake begins to freeze, but it may take several days before it is completely frozen depending on its bathymetry and weather conditions. Moreover, large lakes may not freeze over completely. In consequence, if one wants to correctly represent the ice cover phenology in 1D models, an ice cover fraction has to be
introduced and used to compute a more realistic surface albedo.

Several works have shown the impact of the incorrect representation of lake freezing processes in land surface models (LSMs), e.g., Pietikäinen et al., 2018; Choulga et al., 2019; Garnaud et al., 2022. Based on the Flake lake model, Bernus and Ottlé (2022) also identified shortcomings in the modeling of lake ice phenology in the ORCHIDEE LSM. Although the model evaluation against lake water surface temperatures showed satisfactory results with an RMSE around 2.7 °C at the global scale,
systematic errors in the prediction of ice phenology were highlighted, with too early ice onset and late offset, leading to an overestimation of the ice cover period up to 30 days on average (Bernus and Ottlé, 2022). In a previous study, Bernus et al. (2020) studied the surface temperature sensitivity to model parameters and highlighted the key role of snow and ice radiation coefficients (light extinction and albedo) on ice phenology.

Among these two essential radiative variables, surface albedo is the one that has been mapped at the global scale with remote
sensing for several years. Various satellite products provide daily albedo time series at a few hundred meters scale over the last thirty years which can be used to evaluate and calibrate LSMs. In ORCHIDEE, such products are commonly used to calibrate model parameterizations (Bastrikov et al., personal communication). Recently, Raoult et al. (2023) calibrated the ORCHIDEE snow model over the Greenland ice sheet using MODIS albedo data and showed improvements in the model fit to the observations and the impacts on the simulated snow states, especially the sublimation processes.

In this paper, we present recent work performed in the ORCHIDEE-FLake model to better represent lake freezing through the account of a partial ice cover and the revision of the lake albedo parameterization. The developments are based on the MODIS albedo product which was used both to model ice cover fraction and to calibrate the albedo parameters. The modeled lake ice phenology is evaluated against *in situ* data. Results and following steps for further parameter optimization are finally discussed.

## 2 Data, model and methods




## 2.1 Datasets used

### 2.1.1 HydroLAKES database

The HydroLAKES database identifies all lakes larger than 0.1 km² at the global scale and provides their mean depth (Messager et al., 2016). It was used by Bernus and Ottlé (2022) to generate the lake tiles in the ORCHIDEE grid by clustering the lake area according to their mean depth. In this work, HydroLAKES has been used to mask the studied lakes in the surface albedo time series and to extract the data for the assessment of the ice cover fraction for analyzing its seasonal variations. Data can be downloaded at https://hydrosheds.org/page/hydrolakes (last access: March 2024 ).

### 2.1.2 MODIS data products

We used the MCD43A3 albedo product (Schaaf and Wang (2021), to assess the seasonal evolution of the lake ice cover and to calibrate the model albedo parameters. The data are provided at a daily time step with a 500 m resolution. The bi-hemispherical and directional hemispherical reflectances are given for multiple narrow bands and also for three broadbands (visible, near-infrared, and full solar spectrum corresponding to the surface albedo). The effective surface albedo is a combination of the shortwave bi-hemispherical reflectance also called White Sky Albedo (WSA), and the shortwave directional hemispherical reflectance (Black Sky Albedo, BSA) and depends on the cloud coverage. In the absence of such information, and given that our model does not separate direct and diffuse radiation, we assumed that the average of the retrieved WSA and BSA is a better approximation of the surface albedo as it is modeled in ORCHIDEE. Data are available at https://lpdaac.usgs.gov/products/mcd43a3v006/ (last access: December 2024).

Crossing the albedo raster images with the HydroLAKES polygons allowed us to extract the lake's surface albedo and spatial variability. To avoid radiance contamination by the lakeshore pixels, which are generally covered by vegetation and have larger albedo values compared to water, we used a spectral vegetation index to mask such mixed pixels. The NDVI MODIS product (MOD13A1, Didan, 2015) with the same spatial and temporal resolutions as the MODIS albedo product, was then used for this purpose. Data are available at https://lpdaac.usgs.gov/products/mod13a1v061/ (last access: December 2024).

### 2.1.3 Great Lakes Ice Cover (GLIC) database

The GLIC database provides daily gridded ice cover data at 1.8 km resolution over the Great Lakes from 1973 to present. The dataset fully described by Yang et al. (2020), is based on the standardization of the previously existing GLIC dataset developed by the Canadian Ice Service and the US National Ice Center. Using interpolation techniques (nearest neighbor for space and linear for time), Yang et al. (2020) created coherent data available at the same standardized spatial and temporal resolutions throughout the entire period. The daily data over the 5 lakes (Erie, Huron, Michigan, Ontario, and Superior) can be downloaded at: https://www.glerl.noaa.gov/data/ice/. They were used as a reference to calibrate our albedo-based detection method for the period 2008 – 2018.



### 2.1.4. Lake ice phenology

We used the Global Lake and River Phenology (GLRP) database to evaluate the modeled lake freezing processes (Benson et al., 2000). This dataset provides the onset, offset dates, and duration of observed freezing periods for 857 lakes located worldwide. Data are available for some lakes back to 1850 and up to 2020 and can be downloaded at: https://doi.org/10.7265/N5W66HP8 (last access: March 2024). It should be noted that the reported information is not homogeneous, since, for some lakes, the definition of the ice-on date is the date when the lake is completely ice-covered, while

for the other lakes, it is the date when the lake starts to freeze. We also used the SYKE database, which reports *in situ* ice formation and disappearance for 27 lakes located in Finland, as reported by Choulga et al. (2019).

### 2.1.5. ERA5 meteorological reanalysis

The ERA5 reanalysis dataset was used to simulate the lake thermal processes with ORCHIDEE. This atmospheric forcing is
available globally at an hourly time scale and with a 31 km resolution from 1940 onwards (Hersbach et al., 2020). It was interpolated at a half-hourly time step and a 0.25° resolution, following the methodology presented by Wei et al., 2014. We used the solar and atmospheric downward radiations, the air temperature and humidity, wind speed, precipitation, and atmospheric pressure at the surface to force the model. Data are available at: https://cds.climate.copernicus.eu/cdsapp#!/dataset/reanalysis-era5-complete?tab=overview, last access: March 2024)


### 2.2 ORCHIDEE-FLake model

### 2.2.1 Current status

ORCHIDEE is the continental component of the IPSL Earth System Model which provides the boundary conditions and the energy, water, and carbon fluxes to the atmospheric general circulation model LMDZ (Cheruy et al., 2020). ORCHIDEE can
be run coupled with LMDZ or with prescribed atmospheric conditions delivered by *in situ* measurements or atmospheric forcing datasets. Land surface characteristics are provided by various datasets such as land cover maps which can be updated yearly. In ORCHIDEE, vegetation is described in terms of fifteen Plant Functional Types (PFTs) whose fractions need to be prescribed for each grid point. The model was recently updated with a lake module based on the FLake 1-D lake model (Mironov, 2008), fully described in Bernus and Ottlé, (2022). In the present version, lakes are considered as distinct tiles within
the grid, on which a separate energy budget is solved, independently from the rest of the vegetated grid (no lateral transfers). The lake cover fraction is static and is derived from the HydroLAKES database (Messager et al., 2016), which was used to cluster three categories of lakes according to their average depth. Shallow (depth less than 5 meters), deep (depth over 25 meters), and intermediate lake cover fractions have been derived to generate yearly land cover maps including lake tiles fractions, at various grid resolutions. The bulk energy budgets resolved in FLake allow us to predict the evolution of the vertical
temperature structure within a mixed layer and a thermocline underneath, given several input parameters, namely the lake mean depth, the light extinction coefficient, the water, snow, and ice albedos and the fetch (distance wind blows without



obstruction). In cold conditions, the surface layer can freeze and an ice layer may develop and intercept snow. Because of the 1-D model structure, the whole lake surface freezes when the surface lake temperature falls below 0°C, and the ice thaws above this temperature threshold, regardless of the size of the lake. Water, snow, and ice layer thicknesses and temperatures are prognostic variables calculated at each time step (30 mn in ORCHIDEE).

### 2.2.2 Ice cover fraction

The parameterization proposed by Garnaud et al. (2022) was implemented in ORCHIDEE to represent the lake ice cover, improve the surface albedo, and reduce the cold temperature biases. This model was successfully introduced in the Canadian Small Lake Model (CSLM) and showed improvements in the timing of ice-on/off periods especially for the Great Lakes. Similarly to other parameterizations, such as the one proposed by Choulga et al. (2019) and used in the Integrating Forecasting System of the European Centre for Medium-Range Weather Forecasts (ECMWF), the ice cover fraction is derived from the calculated ice thickness and equals unity above a certain threshold. In Choulga et al. (2022), this threshold is set to 10 cm, which means that above this threshold, the lake tile is completely frozen, and below it, the cover fraction decreases linearly to 0% until the lake is completely thawed. In Garnaud et al. (2022), the threshold is set to a critical value dependent on the wind fetch, to represent the fact that ice is more likely to break under the action of wind stress until it grows to a critical thickness. This critical value $H_{crit}$ may be written:

$$H_{crit} = \frac{\tau_a}{P^*} \times L \tag{1}$$

Where $\tau_a$ is the scale of the surface wind stress (set to 0.15 Pa), $P^*$ is the compressive strength of ice (set to 27.5 kPa) and $L$ is the lake fetch (in meters). The ice cover fraction of the lake tile, *Icefrac,* is then derived from the modeled ice thickness $H_{ice}$ using Eq. 2:

$$Icefrac = \frac{H_{ice}}{H_{crit}} \tag{2}$$

We have implemented this parameterization in ORCHIDEE-FLake as an input to the calculation of the lake surface albedo. For each lake tile, the fetch is static and prescribed to the mean of the fetch of all the lakes falling in the tile. It is estimated at the lake level from the surface extent by assuming a circular shape and taking the diameter of this circle. Given that we did not consider lakes smaller than 0.1 km² due to the limitations of the HydroLAKES database, it means that the fetch values range between a few meters and a few hundred kilometers, leading to critical thicknesses ranging between 2 mm for the smallest lakes to 1.1 m for the larger ones.

### 2.2.3 Albedo revision



In ORCHIDEE-FLake, the lake surface albedo is calculated according to the surface temperature. For free water, the albedo
is set to a value of 0.07 which is the standard value used in FLake. In the presence of ice, possibly covered by snow, the lake
surface albedo ($alb$) depends on the temperature as suggested by Mironov et al., 2012 and varies between two limits
corresponding to wet and dry snow (in presence of snow), and to blue and white ice (if any snow), based on the same equation
(Eq. 3):

$$alb(T_{surf}) = albedo_{max} + (albedo_{min} - albedo_{max})exp\left(\frac{-C_{alb}\times(273.15-T_{surf})}{273.15}\right) \qquad (3)$$

where $albedo_{min}$ and $albedo_{max}$ are respectively the minima and maxima values for ice or snow, $T_{surf}$ is the snow or ice
surface temperature in Kelvin which is always lower than the water freezing point temperature, and $C_{alb}$ a fitted coefficient
equal to 95.6 (Mironov et al., 2012). In Flake, the minimum and maximum albedos are equal to 0.1 and 0.6 respectively, for
both snow and ice. They were revised by Bernus and Ottlé (2022) following Semmler et al. (2012) and Pietikaïnen et al. (2018),
and set to 0.3-0.5 and 0.77-0.87 for ice and snow, respectively (see Table S1 in Supporting Information). In this work, to
account for partial ice coverage, we implemented a supplementary equation, linking linearly the lake albedo to the ice fraction
and accounting for the presence of snow if any. The lake tile surface albedo $alb_{tile}$ is now given by Eq. 4:

$$alb_{tile}(T_{surf}) = Icefrac_{tile} \times alb(T_{surf}) + (1 - Icefrac_{tile}) \times alb_{water} \qquad (4)$$

where $alb_{water}$ is the free water albedo, $alb(T_{surf})$ is the snow albedo in snowy conditions and the ice one otherwise, both
derived from Eq. 3 and dependent on the surface temperature of the lake tile.

**2.3 MODIS Albedo processing on Laurentian Great Lakes**

The MCD43AA albedo product was used to analyze the winter seasonal variations of several lakes for which *in* situ ice cover
observations were available. In particular, the Laurentian Great Lakes, well documented in the GLIC database, and the Finnish
lakes included in the SYKE dataset, were studied. We designed a method to estimate ice cover fractions from the albedo
observations, these fractions were further used to evaluate model parameterizations. Moreover, the albedo data were also used
to calibrate the free water, snow, and ice albedo parameters.

**2.3.1. Ice cover estimation**

The comparison of the lake averaged albedo time series over the 2008-2018 period, with the *in situ* ice cover fractions reported
in GLIC, allowed us to determine a lower threshold of the albedo above which, we could assume that the lake started to freeze.
This threshold appeared to be the same for all five Great Lakes and was found to be 0.15. The maximum value was set to 0.9
since this value was reached for all the lakes that were completely covered once during the studied period. Assuming a linear
relationship between these two extremes, we derived time series of ice cover fractions for each of the Great Lakes. The



comparison of these estimates with *in situ* measurements over the 11 years studied shows a very good agreement, with RMSE
      ranging from 4% for Lake Ontario to 9% for Lake Erié. The latter showed the largest errors because it is the smallest and
      shallowest of the five lakes and has the largest ice fractions.

      We also checked if our methodology could be applied to smaller lakes and tested it on some of the Finnish lakes for which
      Choulga et al. (2019) reported the dates when the lake was completely frozen or completely thawed. Therefore, we compared
these specific dates estimated from the MODIS albedo time series to those observed in the GLIC database and found that over
      the period 2010 - 2015, our method has an average error of 2 days for Lake Nilakka and 5 days for Lake Nasijarvi, the only
      two lakes we could formally identify in the HydroLAKES database.

### 2.3.2. Model parameters calibration

The MODIS albedo time series combined with a literature survey were also used to revise the lake albedo model parameters.
      For free water, even if some of the larger studied lakes showed minimum values around 0.02, most of the lakes present an
      average value of 0.07, which is the value used as a standard in FLake. Lower values, such as the ones observed on the Great
      Lakes (as low as 0.02), could be the result of the larger surface roughness observed over such large lakes, which could increase
      the multiple reflections and reduce the overall surface reflectance. In frozen conditions, the observations confirmed that snow-
covered ice could reach values up to 0.9. Therefore, the maximum values for ice and snow albedos were kept to the prior
      values set in Bernus and Ottlé (2022), i.e., 0.87 and 0.5 for snow and ice respectively. Minimum values being more difficult
      to observe because of the cumulative effect of ice and snow aging and partial coverage, we rely on *in situ* measurements such
      as the ones reported by Svacina et al. (2014), showing snow minimum values close to 0.5, or the ones reported by Lang et al.
      (2018), showing minimum blue ice albedos around 0.15. The prior and posterior values of the albedo model parameters are
summarized in Table S1 in Supporting Information.

### 2.3.2. Model experiments

      The evaluation of the revised albedo parameterization was done at a global scale. Two ORCHIDEE simulations were
      performed: a reference one using the standard model (called "Prior") and another one based on the present developments
(called "Post"), including lake fraction and calibrated albedo parameters. The simulations were performed on the 1979 – 2019
      period and the results were analyzed over the period 2008 – 2018 to ensure a 30-year spinup/warmup period necessary to avoid
      initialization errors. The atmospheric forcing was provided by the ERA5 reanalysis (section 2.1.5) at the half-hourly model
      time step and 0.25° spatial resolution. The lake surface temperatures simulated over each lake tile present in the model grid
      cell were downscaled to extract the lake surface temperatures and ice thicknesses simulated over each lake reported in the
GLRP and the SYKE databases. As a result of missing data over our study period, only 200 lakes among the 857 ones reported
      in the GLRP dataset were used for the model evaluation. The model performances in the representation of ice cover phenology,
      surface albedo and surface temperature are discussed in terms of root mean square errors (RMSE) and improvement factor (or
      error reduction, equal to 1- (RMSE$_{post}$/RMSE$_{prior}$)), with larger values corresponding to larger improvement.



## 3. Results

### 3.1 Lake albedo evaluation on Great Lakes

Figure 1 presents the lake surface albedo simulated by ORCHIDEE over the five Great Lakes for the "prior" and "post" simulations together with satellite-derived albedo observations (MODIS MCD43A3 data). Here, we present daily values averaged over the 2008 – 2018 period to calculate mean seasonal cycles. The prior and post RMSE calculated over the whole freezing period, as well as the improvement factors, are given in Table S2 (Supporting Information).

The results clearly show the improvements brought by the consideration of partial ice cover and the calibration of the minimum values of ice and snow albedos. The prior simulations presented a systematic overestimation of the albedo resulting from the overestimation of the ice cover (equal to 1 during most of the winter season for the five lakes and all the years except the warmer ones i.e. 2012, 2016 and 2017). The RMSEs are considerably reduced with improvement factors larger than 70% (see Table S2, Supporting Information). The error reduction is slightly higher for Lake Michigan, equal to 0.83, and lower for Lake Erie, equal to 0.55. The slightly lower improvement observed for the smallest lake studied is probably linked to the fact that Lake Erie, contrary to the others, can freeze entirely during the colder winters, then reducing the benefit of considering the partial ice cover. Even if the albedo time series do not reveal systematic biases, they show that the albedo can be largely underestimated for some years. This is particularly the case for Lake Erie in 2008 and 2015 or for Lake Huron in 2015 (not shown here). We have seen that for all these lakes, the spatial variability of the observations is very large, demonstrating a large spatial variability of the ice coverage among the lakes. We can also see that the free water albedo is lower in the observations compared to the modeled one (prescribed to 0.07), which leads to systematic positive biases in the albedo simulation.

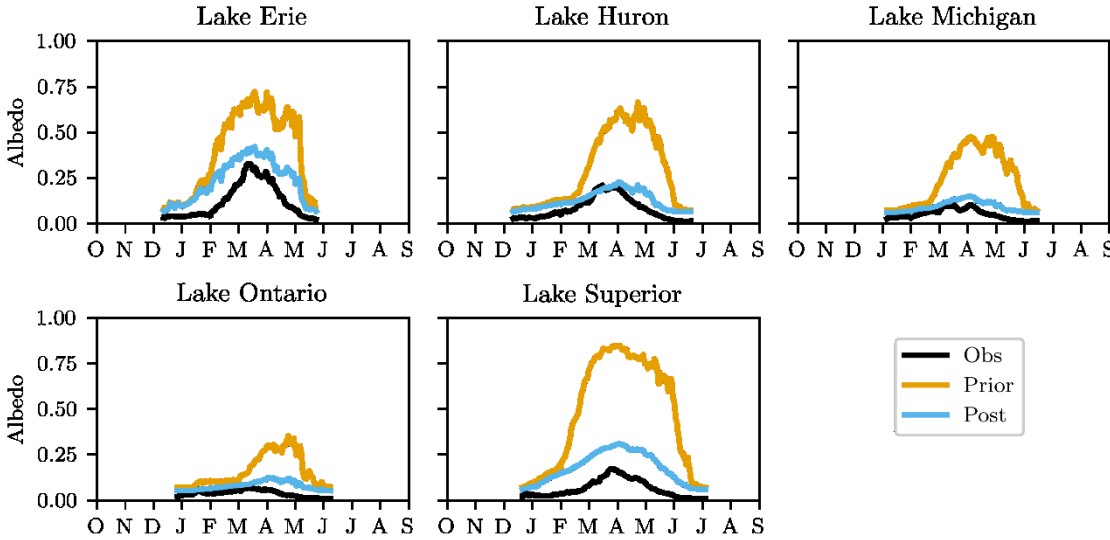



**Figure 1. Lake surface albedo calculated by ORCHIDEE and compared to MODIS albedo observations for the five Great Lakes. Prior refers to the standard model, and Post, to the albedo revised one.**

### 3.2 Evaluation of lake ice phenology

The key dates of the ice cover phenology predicted by ORCHIDEE were compared to the GLIC observations. Figure 2 presents the errors obtained in the simulation of the ice-on, ice-off dates and duration of the ice cover period for the 200 observed lakes, clustered into 3 classes according to their mean depth. In agreement with the observations reported in the GLIC database, the ice fraction was used to diagnose the dates when the lake was completely frozen and ice-free. We assumed that the ice-on and ice-off dates correspond to the dates when the lake is completely frozen and ice-free, respectively. Given that deep lakes

present partial coverage most of the time and that the observations may not be representative of the whole lake, we decided, after various tests, to diagnose the ice phenology dates with the simulated surface temperature as was done in Bernus and Ottlé, 2022. Therefore, the ice-on period starts when the surface temperature falls below 0°C. This approach is empirical and may not be appropriate for all lakes but it allows us to get a first picture of the added value of the fractional ice cover parameterization.

Figure 2 clearly shows how the account for partial freezing in the albedo model improved the timing of the lake freezing, the start and the end of the period compared to the prior simulation, with an overall shift of the date of the start of freezing of 1 day for the shallow lakes and 5 days for the medium and deep lakes. The ice-off also happened sooner by about 9 days for the shallow lakes and by about 12 days for the medium lakes, which reduces the overestimation of the ice cover duration by 12 days in the case of the shallow lakes and by nearly 18 days for the medium ones. The improvements are even more significant

for the deep lakes, especially for the date of the ice-off, which has been brought forward by 18 days. Results obtained for deep lakes show a larger variability of the model errors, especially for the start of freezing which can be explained both by the lower number of lakes sampled compared to shallow and medium lakes (10 compared to 139 shallow lakes and 51 medium lakes), and by the larger uncertainty in the observations given the size of these lakes and the larger spatial variability of the ice coverage. All the results are summarized in Table S3 in Supporting Information.

Besides, the fact that the ice-off date appears to be more impacted by the new parameterization than the ice-on one is explained by the larger sensitivity of the lake energy budget to the surface albedo during the spring months when the ice breaks, compared to the early winter months, when the ice-on period starts for most of the lakes observed. This larger sensitivity is the result of the different amounts of incoming radiation during these two periods, larger in spring compared to winter. Our results also show that the deeper the lake, the larger the impact of the ice fraction on the ice phenology. This was expected, since the length

of the ice-on/ice-off periods increases with lake size and is often positively correlated with depth, explaining the larger time gap between the two ways of parameterizing ice cover fraction and related albedo (Prior vs. Post).





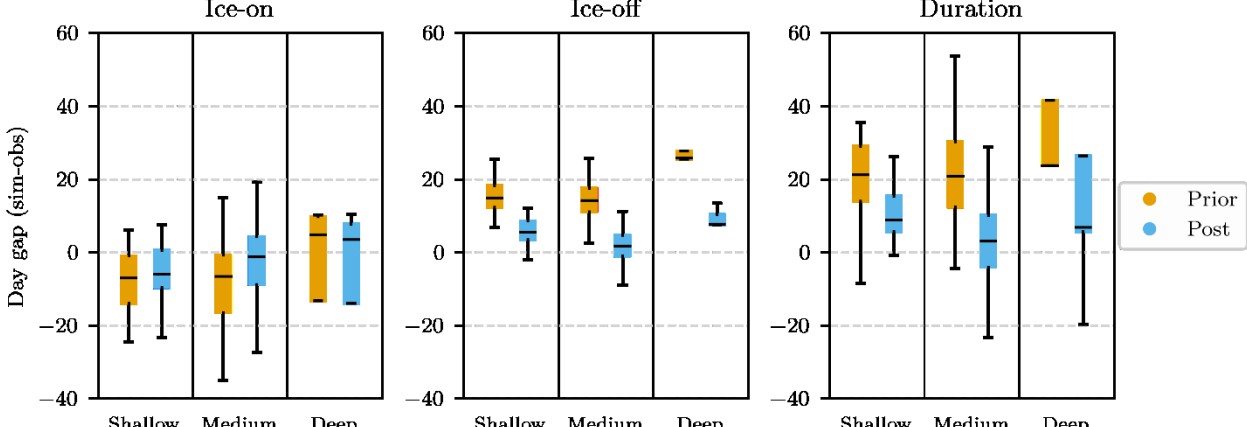

**Figure 2. Ice phenology modeled errors (in days) obtained on the 200 (139 shallow, 51 medium and 10 deep) lakes identified in the GLIC database. The boxplots show the median, first and third quartiles of the time delay (model- observations) as well as the minimum and maximum errors for the ice-on, ice-off and duration of the ice cover period. Prior refers to the standard model, Post to the albedo revised one, time errors are expressed in days.**

## 4. Discussion and Conclusion

The representation of lake freezing processes is crucial in LSMs to correctly simulate the water, energy, and carbon exchanges with the atmosphere. This paper presents the contribution of MODIS albedo observations to calibrate a new representation of the freezing processes in the ORCHIDEE lake module. A methodology was set up to estimate water, snow and ice albedo range of variations, as well as the ice cover fraction from the MODIS albedo time series. A new modeling of the ice fraction, based on Garnaud et al. (2022), was then implemented to better represent the lake surface albedo. Two simulations, with and without accounting for the lake ice fraction, were performed at the global scale to assess the contribution of the new developments on winter lake ice phenology. Our results clearly show the benefits of representing a partial ice coverage on the lake energy budget, for the three depth classes modeled in ORCHIDEE. The ice-on, ice-off and duration of the lake ice cover have been significantly improved, with discrepancies reduced to a few days for the timing of the ice-on and ice-off dates for all categories of lakes (shallow to deep lakes) and a more realistic duration period, with error reductions up to 18 days.

However, the ice-on date is still in advance by about 6 days for shallow lakes, and the ice-off period is too late by about 5 days. This larger residual error for shallow lakes compared to medium ones could be explained by the fact that this category of lakes represents a large proportion of small lakes not explicitly resolved by ORCHIDEE and for which the lake tile mean properties could present larger differences from the real ones compared to the larger medium lakes. Failure to account for this sub-tile variability should have a greater impact on the shallow lake category, for which the depth distribution is the largest. For the deep lakes, we can note a large improvement in the prediction of the ice-off period but still an overestimation of the ice duration by about 7 days.

In this work, we were confronted with the lack of lake ice cover fraction data and, above all, with the lack of homogeneity of these data, with some data documenting the start of the ice period and others the date of complete coverage. In addition, for large lakes, the observations are local and not always representative of the entire lake being modeled. We demonstrated here that remote sensing albedo products are extremely useful for supplementing ice-cover measurements and for monitoring the seasonality of lake ice cover, especially as the surface albedo is crucial for correctly modeling water, energy and carbon balances.

Despite the remaining biases, we think that these developments represent a step forward and pave the way to better simulate the energy and mass exchanges at the atmosphere interface with ORCHIDEE, and especially the water and GHG fluxes essential to study the impacts of climate and environmental changes on lakes. New observational datasets will help us in the development and evaluation of future developments. In particular, the recent satellite-derived products providing global and consistent time series of water levels, extent, ice cover, albedo and surface temperature, such as the ESA – CCI – Lakes project ( https://climate.esa.int/en/projects/lakes/ ), will be essential for our forthcoming works.

**Acknowledgments**

This work was partially supported by the French National Space Agency (Centre National d'Etudes Spatiales) through the TOSCA-SWOT program which funded the Master internship of Z. Titus. The European Horizon project, HORIZON-CL5-2021-D1-01, GreenFeedback (grant agreement 101056921) is also acknowledged for funding A. Cuynet and E. Salmon. The authors would like to thank A. Bernus for transferring model and data processing codes at the start of this work.

**Author contributions:** CO conceived and supervised the study. ZT implemented the new parameterizations, processed the data and performed the simulations. ZT and CO analyzed the results. ZT and AC generated the figures. CO and ZT wrote the manuscript with contributions from all co-authors.

**Competing interests:** The contact author has declared that none of the co-authors has any competing interests.

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
