# Peer review of "Brief communication: Improving lake ice modeling in ORCHIDEE-FLake model using MODIS albedo data"

_EGUsphere, 2024_

## Referee Comment (RC1)

General Comments:

The authors of the paper are utilizing a very commonly used MODIS albedo retrieval and parameterizing this variable into the FLAKE model that is then incorporated into the ORCHIDEE land surface model. The aim of this manuscript is to highlight that by parameterizing the albedo retrievals from MODIS that it better constrains the albedo parameter of the lake ice thickness model, thereby improving the ice-on and ice-off dates, providing a better assessment of ice phenology for lakes of various sizes, from small up to the Laurentian Great Lakes. In my assessment, the authors have done well to incorporate the MODIS data into the FLAKE model used in the ORCHIDEE land surface model, and I do believe that the improvements they are reporting are conceivable.

The presentation and assessment of the outputs of the model, in the Observed/Prior/Post experiment is lacking in detail, glossing over much of the potential detail that could have been provided for both the Great Lakes GLIC dataset, and the GLRP database. There needs to be significant work done to a) show the inter-year distribution of albedo parameter improvement in the Great Lakes dataset, and b) show how the improvement of ice phenology parameters varied across distance and time. The authors do well to describe the spatial distribution of lakes within the study among multiple countries, but do not present a study site figure, or segment the results by administrative or geographical boundaries (which have influences on how and the frequency of phenology reports). Section 3.1. presents a significant improvement to the model since the albedo is being parameterized, but it does not read as an important contribution, mainly because it is only shown through an average of 10 years for each lake. More detail in the inter-year distribution of results needs to be shown.

With improvement to the presentation of the study site, methods, and results section this manuscript would be an important contribution to the body of literature.

Specific Comments:

Page 1 Line 19 – 20: "The role of lakes… demonstrated in various works"

*Briefly describe what the role of lakes actually are, not that other people say they're important.*

Page 2 Lines 37-38: "In these large-scale models… linked to the varying bathymetry"

*Please provide references for this statement.*

Page 2 Line 44: "free water and ice show very different characteristics"

*In what way? Please be specific*

Page 2 Lines 47 – 48: "When the air temperature falls… on its bathymetry and weather conditions"

*This is an oversimplification of how freezing occurs – what about general density of water at 4 degrees, and the mixing that happens?*

Page 2 Line 60: "daily albedo time series at a few hundred metres"

*Be specific- how many metres?*

Page 3 Line 89: "Crossing the albedo raster images"

*Do you mean "overlaying"?*

Page 3 Lines 90 – 91: "To avoid radiance contamination by the lake shore pixels... to mask such mixed pixels".

*This is a surprising way to mask out all non-water pixels. For instance, what about exposed soil/sand? Those pixels would not have vegetation but would have higher albedo than water, so it could still erroneously be included for small and medium lakes.*

Page 3 Line 100 – 102: *The GLIC provides a slew of ice concentration data – but how was it derived? It's important to note that it was derived using a combination of paper charts, SAR, visible/infrared imagery and met data, not just interpolated.*

Page 4 Line 110: "SYKE"

*Define.*

Page 4 Line 124: "LMDZ"

*Define.*

Page 7: *There is no workflow diagram shown here – which is necessary to help the reader understand the new implementation and model experiment.*

Page 7 Lines 206 – 207: "the only two lakes we could formally identify in the HydroLAKES database"

*This needs a better explanation. Are these the only two lakes with names? How does one identify the lakes?*

Page 7 Lines 213 – 214: larger surface roughness observed... and reduce the overall surface reflectance"

*Would the surface roughness increases not actually increase reflectance as well? Surface roughness on water could cause white caps?*

Page 8 Line 238 – 239: "Here, we present daily values averaged over the 20098-2018 period..."

*There are a lot of dates averaged together here, and one year may be very different from another, especially considering the size of the Great Lakes. There is no presentation of the standard deviation of the albedo values, the interquartile range of the deviation of the albedo and no presentation of the spread of observations, prior or post. The results that are presented are a significant wrapping up of the data, presenting 10 years in one line.*

Page 8 Line 242: "The results clearly show"

*Conjecture, please avoid using the word "clearly".*

Page 9 Lines 244 – 245: "The RMSEs are considerably reduced... Supporting Information)/.

*Some of this needs to be included in the manuscript – it's important for the reader to see.*

Page 8 Line 247: "then reducing"

*Do you mean "thus reducing"?*

Page 8 Line 250: "not shown here)"

*When highlighting a particular year to discuss deviation in expectations of results, this needs to be shown.*

Page 8 Line 250 – 251: We have seen that for al these lakes, the spatial variability is very large..."

*How can this be shown within the manuscript?*

Page 9 Line 262: "according to their mean depth"

*Reminder the reader what shallow, medium and deep mean here*

Page 9 Lines 264 – 269: "Given that deep lakes... fractional ice cover parameterization"

*This needs to be included in the methods, not introduced in the results.*

Page 9 Line 270: "Figure 2 clearly..."

*Conjecture*

Page 9 Line 271 – 279: *This description of the data would be well served in a table that is included in the manuscript.*

Page 9 Line 280 – 283: "Besides, the fact that the ice-off... most of the lakes observed"

*Is this not more likely because the albedo changes significantly quickly in the spring compared to the Fall, where ice onset may be congelation ice with albedo increasing less quickly?*

Figure 2: *Are there values for data representing one year? Multiple years? Is there more or less agreement based on the year observed? What about the RMSE of the sim-observed values? There are a lot of potential tests that can be run to showcase how the correction is behaving across space and time. Also, GLRP is worldwide, you could segment the data across administrative or geographic boundaries.*

Figure 2 caption, Line 289: "GLIC database"

*Is this not the GLRP database?*

---

## Author Comment (AC1)

Response to Reviewer 2:

We thank the anonymous reviewer 2 for taking the time to proofread and provide insightful comments on the manuscript. We have done our best to address all comments in the revised version to improve the overall quality of the manuscript in line with the reviewer's recommendations. Our responses are reported below in blue.

The Flake lake model within the ORCHIDEE land surface model has been updated by the Authors to better simulate winter ice cover. Using MODIS albedo data and the Great Lakes Ice Cover fraction dataset, the authors calibrated and validated a new lake albedo parameterization that accounts for partial ice cover. This update significantly improved the simulation of ice phenology for 200 lakes of various sizes, as reported in the Global Lake and River Phenology database. The improvements were notable across all lake sizes, with the largest and deepest lakes showing the greatest reduction in error for ice cover duration. The study underscores the importance of considering partial ice cover to accurately model lake ice.

Overall, the work presents a useful contribution to large-scale modelling in partially ice-covered regions but does leave some lack of clarity on how this performs in fully ice-covered regions (e.g. by implying Lake Erie did not improve as much as the rest due to the more uniform ice cover). Are the 200 lakes this is tested on in the warmer regions that have partial ice cover? Or the full GLRP, which includes many northern lakes with full ice cover? And are the SYKE lakes in southern Finland? Or in the north with solid ice covers? While acknowledging this is a brief communication, some important details have been omitted, and once addressed, this paper could be considered for publication.

Thanks for your constructive remarks. We agree that the GLRP and the SYKE databases are not well described. We added more details in the revised version and added a figure (figure S3) in the supplementary document to map the 200 lakes that are used in the evaluation of the lake ice phenology. Because of missing data over our study period (2008-2018), only 200 lakes among the 857 reported in GLRP could be used for the evaluation. These lakes are mainly located in the northeastern part of the US and Canada and in Scandinavia. Even if their location does not cover the whole circumpolar region, they cover a large diversity of atmospheric conditions with lakes presenting a complete ice cover during the cold season to lakes never freezing entirely.

Line 73: Hydrolakes – While agreed that it is the best option available, the limitations in lake depths should be noted here as outlined in Messager et al., 2016 (estimating the depths based on limited information in some locations), and what the implications are for your study.

We agree that the mean depth provided by HydroLakes is subject to large uncertainties. In order to better address these limitations, we added a sentence in the presentation of the database to explain how the lake depth was estimated and the model errors linked to

depth errors are also mentioned in the discussion section. See Lines 73-74 and Lines 315-317 of the revised document.

Line 90: Perhaps comment on why grouping the lake into similar depth tiles rather than just run for each 500m pixel? What if there is fractional ice coverage within each depth grouping?

In this work, we wanted to evaluate the ORCHIDEE-FLake model as it was developed for the modeling of the energy, water and carbon fluxes at the global scale. In this version, the model is forced with atmospheric reanalysis at 0.25° resolution. At such a large scale, lakes for the most part, are not represented explicitly and only lake tiles or effective lakes are represented in the horizontal model grid (as it is done for the vegetation). We have chosen to consider only three effective lakes differing by their depth in each grid cell. Effective depth is given by the mean average of all lakes located in the grid cell, clustered in three categories (shallow, medium and deep). The model evaluation is done by comparing the observed lake to its corresponding effective lake in the grid. We agree that our methodology was not clear in the first version of our paper. We revised therefore the presentation of the model (see Lines 146-152) and the Model experiments section (Lines 248 to 263)

Also is the MODIS water mask not a viable option rather than using a vegetation index?

The MODIS water mask is static (Feb 2000) and based only on the SRTM water mask, based on surface elevation data. Such data can be used to map surface water but don't indicate if the water is covered by aquatic vegetation which has a very different albedo compared to water. This is why we tried to use NDVI data to remove water vegetation, to assess the free water albedo. This has been done in the earlier phase of the work but was finally not kept further. Therefore, we decided to remove this point in the description of the albedo processing. See the slight modifications in section 2.1.2.

Line 132: is the medium category is missing here?

No, this is a mistake from our side; the medium category is the same as the intermediate category. We corrected the word in Line 127.

Line 180: "In Flake, the minimum and maximum albedos are equal to 0.1 and 0.6 respectively, for both snow and ice. They were revised by Bernus and Ottlé (2022) following Semmler et al. (2012) and Pietikaïnen et al. (2018), and set to 0.3-0.5 and 0.77-0.87"

These ice albedos are a lot lower than what have been published for lakes in Central Ontario. Are those albedo values to represent a broken ice cover only? or only during the melt season?

The value of 0.1 for ice is the one prescribed in FLake model (see documentation and related publication). Compared to what has been previously published, we also found that they were too low; this is why in our previous work referenced here (Bernus and Ottlé,

2022) it was set to 0.3. In this work, we observed values as low as 0.15 for dates where ice cover was still reported to be present. It was indeed during the melting period. This is why we decided to reduce the minimum value of the ice albedo to 0.15.

~ line 200 – I think it would be pertinent to show the results here. Also, add more explanation – if Lake Erie has the largest ice fraction, and the worst RMSE does that not indicate that your new method is not suitable for lakes with a full ice cover and should only be used on a partial ice cover? Or is the albedo selected based on the ice cover fraction? This needs some clarity.

Because of the format of the brief communication, we are limited in the number of figures and tables. We chose not to add too many details on the methodology that we have developed to assess the relationship between ice coverage and albedo and to focus our paper on the results obtained after implementation and calibration of the ice cover fraction into ORCHIDEE-FLake. All the results are available in the Master thesis of Z. Titus (in French) which could be referenced if you think it necessary.

Line 205: "Therefore, we compared these specific dates estimated from the MODIS albedo time series to those observed in the GLIC database and found that over the period 2010 - 2015, our method has an average error of 2 days for Lake Nilakka and 5 days for Lake Nasijarvi, the only two lakes we could formally identify in the HydroLAKES database."

Unclear – why are there Finnish lakes in the GLIC database? Clarify why only the 5 years?  Can you not use maps to identify the other lakes in the database? The explanation here needs more clarity. And again, does this indicate that these lakes had partial ice cover? Or did the results improve for full ice cover here, but not for Lake Erie?

We are sorry, we meant the SYKE database and not the GLIC database. The GLIC data concerns only the Laurentian Great Lakes. Sorry for the confusion. It is corrected in the revised version.

Line 230:  which lakes? Where?  From what I recall the GLRP is not really global, it's mostly North American - are your test lakes mainly in Canada then? in regions with full ice covers?  The work needs a map to show which lakes you are using.

We followed your suggestion and added a map as a Supplementary figure (Figure S3) to show the GLRP lakes used for the model validation (see also response to the general comments above.

Line 249: "This is particularly the case for Lake Erie in 2008 and 2015 or for Lake Huron in 2015 (not shown here)."

Consider showing the examples and why the warm/cold year performs well/not well.  Perhaps in the supplemental section if length is an issue for the brief communication. Variability is important.

We fully agree that time variability is very important and finally replaced Figure 1 with a new figure showing the interannual variability and model performances over the studied period (2008-2018). Moreover, we added as supplementary Figure S2 the prior and remaining model errors for the five Great Lakes, for each studied year, highlighting with a specific color scale, the degree of coldness of the winter, from the warmest in dark red to the coldest in dark blue. We tried to better discuss in the results section 3.1, the performances of the model according to the atmospheric temperature conditions. We also transfer the previous Figure 1 showing the interannual mean average of the surface albedo to Figure S1 (supplementary document), in order for the reader to get a global picture of the model improvements. Finally, we added Table S3 in supplementary, to give more quantitative results of the comparison, with minimum, maximum, standard deviation and interquartile range of the annual distributions of the water albedo, observed and modeled. We hope that these changes make it easier to appreciate the added value of our developments.

Line 265: "Given that deep lakes present partial coverage most of the time and that the observations may not be representative of the whole lake, we decided, after various tests, to diagnose the ice phenology dates with the simulated surface temperature as was done in Bernus and Ottlé, 2022."

Which deep lakes have partial coverage? Where? How far north? this is something that needs to be clearly addressed throughout.

In ORCHIDEE-FLake, the effective lakes are clustered in 3 categories (shallow, medium and deep) according to the HydroLakes database as it is hopefully, better explained in Section 2.2.1 in the revised version. We chose to diagnose the start and end of the freezing season differently for the modeled deep lakes, based on the surface water temperature since we realized that the observed dates are local and are not representative of the whole lake. For the shallow and medium categories, the full ice coverage is generally observed and can be compared to the simulated ones. For the deep lakes (which represent only 10 lakes over the 200 simulated), the start and end of freezing are diagnosed in the model with the water temperature and the 0°C threshold. We better explain our methodology in the Methods section (Model experiments) and added this uncertainty issue in the discussion.

Figure 2: Do you mean the global river and lake ice database? Not the great lakes ice cover?

We are sorry, there was a typo in the legend of Figure 2. The data are coming from the GLRP database and not the GLIC one. We corrected this error in the revised version. Thanks again for your careful reading.

---

## Author Comment (AC2)

Response to Reviewer 1:

We would like to thank the anonymous reviewer 1 for taking the time to proofread and provide insightful comments on the manuscript. We have done our best to address all comments in the revised version so as to improve the overall quality of the manuscript in line with the reviewer's recommendations. Our responses are reported below in blue.

General Comments:

The authors of the paper are utilizing a very commonly used MODIS albedo retrieval and parameterizing this variable into the FLAKE model that is then incorporated into the ORCHIDEE land surface model. The aim of this manuscript is to highlight that by parameterizing the albedo retrievals from MODIS that it better constrains the albedo parameter of the lake ice thickness model, thereby improving the ice-on and ice-off dates, providing a better assessment of ice phenology for lakes of various sizes, from small up to the Laurentian Great Lakes. In my assessment, the authors have done well to incorporate the MODIS data into the FLAKE model used in the ORCHIDEE land surface model, and I do believe that the improvements they are reporting are conceivable.

The presentation and assessment of the outputs of the model, in the Observed/Prior/Post experiment is lacking in detail, glossing over much of the potential detail that could have been provided for both the Great Lakes GLIC dataset, and the GLRP database. There needs to be significant work done to a) show the inter-year distribution of albedo parameter improvement in the Great Lakes dataset, and b) show how the improvement of ice phenology parameters varied across distance and time. The authors do well to describe the spatial distribution of lakes within the study among multiple countries, but do not present a study site figure, or segment the results by administrative or geographical boundaries (which have influences on how and the frequency of phenology reports). Section 3.1. presents a significant improvement to the model since the albedo is being parameterized, but it does not read as an important contribution, mainly because it is only shown through an average of 10 years for each lake. More detail in the inter-year distribution of results needs to be shown.

We are sorry for this misunderstanding. We don't represent explicitly all the studied lakes in ORCHIDEE-FLake. The model is run at a 0.25° resolution and we represent in each grid cell, three categories of lakes (shallow, medium and deep). These effective lakes may be compared to actual lakes falling in the same category. For the larger lakes occupying the entire grid cell, there is no ambiguity and the relationship is univocal. If there is more than one lake in a given category, the comparison is complicated since the median conditions of the effective lake may differ from the observed actual lake. We have tried to better explain our lake representation in ORCHIDEE in the revised version and we added a discussion on the model uncertainties in the discussion/conclusion section.

With improvement to the presentation of the study site, methods, and results section this manuscript would be an important contribution to the body of literature.

Specific Comments:

Page 1 Line 19 – 20: "The role of lakes... demonstrated in various works"

Briefly describe what the role of lakes actually are, not that other people say they're important.

The role of lakes was described later in the paragraph but we agree that this first sentence could be improved to better introduce our work. Therefore, the first paragraph was changed in the revised version and we hope that our subject is now better introduced. See lines 19 to 31.

Page 2 Lines 37-38: "In these large-scale models... linked to the varying bathymetry"

Please provide references for this statement.

Page 2 Line 44: "free water and ice show very different characteristics"

In what way? Please be specific

Page 2 Lines 47 – 48: "When the air temperature falls... on its bathymetry and weather conditions"

This is an oversimplification of how freezing occurs – what about general density of water at 4 degrees, and the mixing that happens?

We agree that this paragraph contained generalities and oversimplifications and we completely revised it. The format of the brief communication does not allow us to enter into details but in the revised version, we have tried to remain synthetic while being rigorous. We hope that the new version answers to your comment.

Page 2 Line 60: "daily albedo time series at a few hundred metres"  Be specific - how many metres?

There are different albedo products available in the scientific community at different temporal and spatial scales. If we consider only the medium-resolution albedo products, the spatial scales may vary between 300 m to 1 km. Here we worked with a MODIS 500 m product which is described in the data section (2.1.2). In this general introduction, we think

that we can stay general given the large amount of data providers, given that we describe the product used in more detail later.

Page 3 Line 89: "Crossing the albedo raster images"  Do you mean "overlaying"?

YES, that is what we meant. The word was changed in the revised version.

Page 3 Lines 90 – 91: "To avoid radiance contamination by the lake shore pixels... to mask such                                          mixed                                          pixels".

This is a surprising way to mask out all non-water pixels. For instance, what about exposed soil/sand? Those pixels would not have vegetation but would have higher albedo than water, so it could still erroneously be included for small and medium lakes.

We agree that our presentation was not clear. In fact, we were considering only the filtering of the lake's aquatic vegetation which can modify greatly the lake albedo and that we encountered when working on tropical lakes. Given that the paper is focused on cold regions, we realized that this filtering was not effective in our processing given that aquatic vegetation is not an issue there. So we remove this sentence and the presentation of the NDVI product in the revised version. Sorry for this left-over of a report on the albedo optimization at the global scale.

Page 3 Line 100 – 102: The GLIC provides a slew of ice concentration data – but how was it derived? It's important to note that it was derived using a combination of paper charts, SAR, visible/infrared imagery and met data, not just interpolated.

Sorry for the shortcuts. We agree with your corrections and modified the description of the GLIC database in the revised version (see lines 92 to 98).

Page 4 Line 110: "SYKE"   Define.

Done in the revised version

Page 4 Line 124: "LMDZ"     Define.

This name can not be better defined since it is not an acronym but the name of the model. For your information, historically, the atmospheric model was developed by the Laboratoire de Meteorologie Dynamique in France, and Z was referring to the zoom capacity of one version. But now, it is a community model, the original name was kept, but it is no longer an acronym of a model version. So we should leave it like it is written, as it is done in all the papers/works using this model.

Page 7: There is no workflow diagram shown here – which is necessary to help the reader understand the new implementation and model experiment.

The format of a brief communication does not allow us to enter into the details of the model workflow. The ORCHIDEE-FLake model developments are described in Bernus and Ottlé, 2022. Here, we have chosen to only present the parameterizations which were modified. We have revised the methodology section in various places to make it clearer. We hope that it will be enough to better understand the developments and the performed experiments.

Page 7 Lines 206 – 207: "the only two lakes we could formally identify in the HydroLAKES database". This needs a better explanation. Are these the only two lakes with names? How does one identify the lakes?

We modified the sentence to make it clearer. These 2 lakes are indeed the only ones that we could identify in HydroLAKES and then simulate with our model. We could not identify the others from their coordinates in SYKE, unfortunately.

Page 7 Lines 213 – 214: larger surface roughness observed... and reduce the overall surface reflectance"

Would the surface roughness increases not actually increase reflectance as well? Surface roughness on water could cause white caps?

We agree that wind roughness could cause white caps at the local scale and modify the reflectance in the visible. Here, we are considering the overall solar spectrum and the large-scale albedo averaged over the whole lake. We assume that at this scale, the multiple reflections/transmissions of the solar radiation are larger when the surface is rough compared to a smooth one and that it will result in less radiation transmitted upward, therefore a lower albedo. We agree that it is an assumption but since it seems to be specific to large lakes and not observed over smaller ones, it can not be a sensor calibration issue, so it should have a physical explanation. We therefore propose to put forward this explanation in the revised paper.

Page 8 Line 238 – 239: "Here, we present daily values averaged over the 20098-2018 period..."

There are a lot of dates averaged together here, and one year may be very different from another, especially considering the size of the Great Lakes. There is no presentation of the standard deviation of the albedo values, the interquartile range of the deviation of the albedo and no presentation of the spread of observations, prior or post. The results that are presented are a significant wrapping up of the data, presenting 10 years in one line.

We thank you for your constructive suggestions, therefore we replaced Figure 1 with a new one showing the annual daily time series over the whole period. We moved the former Figure 1 in the supplementary document and added as well a new supplementary Figure S2 presenting the absolute errors obtained with the prior and the new model for the five Great Lakes differentiating each year with a color scale related to the coldness of the winter season, from the coldest in dark blue to the warmest in dark red. Moreover, we added Table S3 in supplementary, to give more quantitative results of the comparison, with minimum, maximum, standard deviation and interquartile range of the annual distributions of the water albedo, observed and modeled. We hope that these changes make it easier to appreciate the added value of our developments.

Page 8 Line 242: "The results clearly show"  Conjecture, please avoid using the word "clearly".

Corrected

Page 9 Lines 244 – 245: "The RMSEs are considerably reduced... Supporting Information)/.

Some of this needs to be included in the manuscript – it's important for the reader to see.

We agreed and added the ranges of the RMSE values for the two model versions in the main text. See Lines 244 - 246.

Page 8 Line 247: "then reducing"  Do you mean "thus reducing"?

Yes, it was corrected.

Page 8 Line 250: "not shown here)"  When highlighting a particular year to discuss deviation in expectations of results, this needs to be shown.

This is now done in the new version since Figure 1 now presents the daily time series.

Page 8 Line 250 – 251: We have seen that for al these lakes, the spatial variability is very large..."

How can this be shown within the manuscript?

As we explained above, we changed Figure 1 and added supplementary Figures S1 and S2 as well as Table S3 to better present and easily follow the discussion.

Page        9        Line        262:        "according        to        their        mean        depth"

Reminder the reader what shallow, medium and deep mean here

Done

Page 9 Lines 264 – 269: "Given that deep lakes… fractional ice cover parameterization"

This needs to be included in the methods, not introduced in the results.

We agree with your suggestion and move this part to the method section, see lines 225-230.

Page           9           Line           270:           "Figure           2           clearly…"

Conjecture

Corrected

Page 9 Line 271 – 279: This description of the data would be well served in a table that is included in the manuscript.

Given the limitations of the letter format, we could not add a Table in the main manuscript. We tried to better explain the model and experiment protocol in the new version and hope that it will be enough. Otherwise, we could add a supplementary table.

Page 9 Line 280 – 283: "Besides, the fact that the ice-off… most of the lakes observed"

Is this not more likely because the albedo changes significantly quickly in the spring compared to the Fall, where ice onset may be congelation ice with albedo increasing less quickly?

This is related indeed. Therefore, we changed slightly the sentence in the revised version in order to insist on the fact that the freezing/melting processes have different dynamics and durations.

Figure 2: Are there values for data representing one year? Multiple years? Is there more or less agreement based on the year observed? What about the RMSE of the sim-observed values? There are a lot of potential tests that can be run to showcase how the correction is behaving across space and time.  Also, GLRP is worldwide, you could segment the data across administrative or geographic boundaries.

Figure 2 represents the distribution of the errors calculated for each year and for all the lakes of each lake category. The boxplot shows the median, first and third quartiles of these errors. It is difficult to analyze the spatial distribution of these errors since the studied lakes do not present a homogeneous distribution all over the world. We added Figure S3 to

highlight their location. Concerning the temporal distribution of the errors, Figure 2 and Table S4 show that the errors for the ice-on and the ice-off dates have about the same order of magnitude and new Figure S2 highlights that the errors are larger during the coldest winters compared to the warmest ones, for the Laurentian lakes. The relationship with the coldness of the winter is difficult to analyze for the 200 lakes since the weather variability is spatially different. We hope that the new figures and discussions make this point clearer in the revised version.

Figure 2 caption, Line 289: "GLIC database" Is this not the GLRP database?

Yes, sorry for this mistake, we corrected it in the revised version. Thank you again for your constructive suggestions and your careful reading.